# ssDNA recombineering boosts in vivo evolution of nanobodies displayed on bacterial surfaces

Yamal Al-ramahi [1], Akos Nyerges [2,4], Yago Margolles[3], Lidia Cerdán [3], Gyorgyi Ferenc[2], Csaba Pál [2], Luis Ángel Fernández [3✉] & Víctor de Lorenzo [1✉]

ssDNA recombineering has been exploited to hyperdiversify genomically-encoded nanobodies displayed on the surface of *Escherichia coli* for originating new binding properties. As a proof-of-principle a nanobody recognizing the antigen TirM from enterohaemorrhagic *E. coli* (EHEC) was evolved towards the otherwise not recognized TirM antigen from enteropathogenic *E. coli* (EPEC). To this end, *E. coli* cells displaying this nanobody fused to the intimin outer membrane-bound domain were subjected to multiple rounds of mutagenic oligonucleotide recombineering targeting the complementarity determining regions (CDRs) of the cognate VHH gene sequence. Binders to the EPEC-TirM were selected upon immunomagnetic capture of bacteria bearing active variants and nanobodies identified with a new ability to strongly bind the new antigen. The results highlight the power of combining evolutionary properties of bacteria in vivo with oligonucleotide synthesis in vitro for the sake of focusing diversification to specific segments of a gene (or protein thereof) of interest.

[1] Systems and Synthetic Biology Department, Centro Nacional de Biotecnología (CNB-CSIC), Campus de Cantoblanco, Madrid 28049, Spain. [2] Synthetic and Systems Biology Unit, Institute of Biochemistry, Biological Research Centre, Szeged H-6726, Hungary. [3] Department of Microbial Biotechnology, Centro Nacional de Biotecnología (CNB-CSIC), Campus de Cantoblanco, Madrid 28049, Spain. [4] Present address: Department of Genetics, Harvard Medical School, Boston, MA 02115, USA. ✉email: lafdez@cnb.csic.es; vdlorenzo@cnb.csic.es

Antibodies are the best instance of functional biomolecules which owe their performance to the ability of their encoding DNA of combining in vivo a limited number of hypervariable sequences with a virtually fixed protein scaffold[1,2]. While the variable (V) domain of different antibodies adopt a very similar immunoglobulin (Ig) fold structure, their antigen-binding loops (the so-called complementarity determinant regions or CDRs) are selected out of a very large diversity pool[3]. Such a process occurs upon exposure of the immune system to given antigens through an intricate molecular mechanism of clonal selection, amplification and affinity/specificity maturation[4,5]. One remarkable feature of this course is that in vivo diversification of the CDR sequences for optimally binding a molecular target is virtually limited to 2–4 residues of the primary protein sequence[5]. The single V domain of the naturally occurring heavy chain camel antibodies—also called VHHs or nanobodies (Nbs) when expressed in a recombinant form—represent an extreme case of this state of affairs[6]. While the structures of the Ig V-domain scaffold of nanobodies are virtually identical, their CDRs (Fig. 1) are highly variable, both in length and amino acid composition[7].

Given the structural simplicity of camel-derived Nbs it comes as little surprise that they have emerged as a favourite platform for recreating in the laboratory some of the mechanisms involved in their natural diversification and thus breed large libraries which can later be panned for selecting best binders to specific targets. In particular, the possibility to express VHHs in multiple surface display platforms[8] (i.e. phages, yeast, bacteria) or ribosome display[9] systems enables the selection of variants after diversification. Variability can be introduced either in vivo with a limited targeting of the VHH sequence[10] or in vitro with degenerate codons at CDRs using PCR with oligonucleotide primers[11] or DNA synthesis[12,13]. Yet, while such in vitro methods for creating targeted variability in the CDRs push the potential number of antibody variants close to those of the natural immune system ($>10^{13}$)[14] they are limited to in vitro selection methods (e.g. ribosome display) that avoid cloning and cell transformation

steps needed for more robust and versatile in vivo selection approaches (e.g. phage, bacterial or yeast display)[15].

On this background we wondered whether we could boost a targeted diversification of the CDRs of VHHs by combining the evolutionary power of bacteria in vivo with the ease of synthesizing mutagenic oligonucleotides in vitro. Compared to other methods, oligonucleotide-mediated in vivo mutagenesis (i.e. oligo recombineering) can generate extreme diversity (i.e., $>10^{10}$ variants/library) with single nucleotide precision in living cells. Furthermore, as such oligo recombineering can be iterated and coupled to continuous selection, it allows the continuous generation and selection of evolved variants with beneficial mutations[16]. We thus envisioned a combination of surface-display of single-copy genome-encoded Nbs on E. coli cells[17] with the cyclic and DNA-segment directed mutagenesis method called DIvERGE[18]. This in vivo ssDNA recombineering strategy progressively hyper-mutagenizes specified segments of any gene of interest (e.g. the CDRs of a given VHH) without off-target mutagenesis in adjacent DNA. This could then be followed by capturing by physical means the bacteria displaying the best nanobody variants on their surface[8,19].

In this article we show how combination of ssDNA recombineering[16] with surface-displayed nanobodies on E. coli cells enables rapid expansion of the antigen targeting capacity of a Nb that binds the extracellular domain of the translocated intimin receptor (TirM) of enterohaemorrhagic E. coli (EHEC) strains towards the otherwise non-recognized but functionally homologous TirM domain of enteropathogenic E. coli (EPEC) strains[20,21]. The resulting, evolved nanobodies recognizing the new TirM$^{EPEC}$ antigen were then secreted and easily purified from bacterial culture media by adopting the E. coli hemolysin protein secretion system[22] integrated in the experimental pipeline. In this way, the work below results in nanobodies potentially useful for diagnostics and passive immunotherapies against EPEC and EHEC infections causing human diarrhoeal diseases[20,23]. Furthermore, the data below provides an example of how an all-prokaryotic platform can be designed to artificially embody key features of the immune system for a large number of applications —including diagnostics, therapeutics and environmental needs e.g. detection of biodegradative enzymes as pollution markers[24].

## Results and discussion

**Rationale for ssDNA recombineering-based diversification of anti-TirM Nb TD4.** The Nb called TD4 has high affinity ($K_D < 5$ nM) and specificity for TirM$^{EHEC}$[21], a protein domain required for intimate attachment of the pathogen to intestinal epithelial cells with actin pedestal formation during infection by EHEC[25,26]. However, TD4 Nb did not show any significant binding for the equivalent TirM antigen of EPEC strains, despite the fact that the 24-amino acid epitope recognized by TD4 is 71% identical in both variants[21]. On this basis, we picked TD4 as a good candidate for sampling whether one given Nb could be evolved with DIvERGE technology[18] towards new binding properties. To this end, we adopted E. coli strain EcM1luxSAtir[17] (Supplementary Table S1) as the primary experimental platform for V$_{HH}$ diversification. This strain (Fig. 1) has a chromosomal insertion of a genetic construct encoding the TD4 Nb fused to the C-terminus of the outer membrane anchoring domain of intimin[17]. This fusion is expressed through the constitutive P$_{N25}$ promoter[27] and it is integrated replacing the flu (Ag43) locus of E. coli K12 genome[28]. Furthermore, EcM1luxSAtir strain was deleted of the fimA-H operon, encoding type 1 fimbriae[29] and has an insertion of the luxCDABE operon of Photorhabdus luminescens[30] into the matBCDEF operon, encoding the E. coli common pilus[31]. These genetic modifications inactivate the expression of cognate

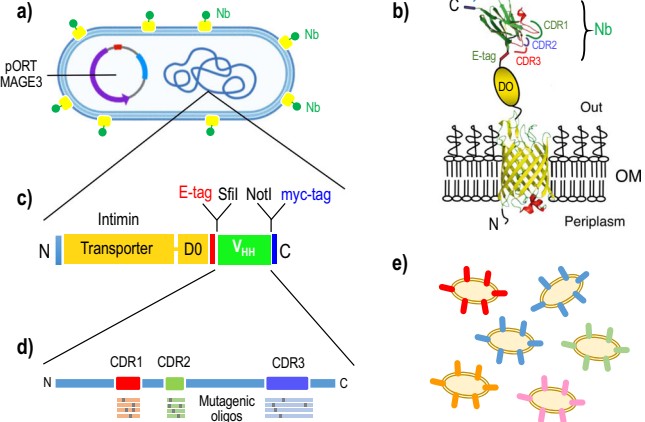

**Fig. 1 A new platform for the generation and screening of VHH variants with new antigen binding specificities. a** The E. coli bacteria used in this work are transformed with plasmid pORTMAGE3 (Supplementary Table S2) and display on their surface **b** a fusion between an outer membrane (OM)-bound intimin domain and a Nb (Nb) which holds the 3 complementarity determinant regions (CDRs) for antigen binding flanked by diagnostic epitopes E-tag and myc as indicated. The cognate DNA sequences are encoded in the chromosome (**c**), where they can be targeted by mutagenic oligonucleotides (**d**) and generate a population with an increased diversity in the sequences of the CDRs thereby presented on the surface of individual bacteria (**e**).

endogenous *E. coli* cell surface structures that could interfere with binding of the displayed Nb to target antigens[17]. This makes *E. coli* EcM1*lux*SAtir an optimal strain for displaying the Nb moiety of the fusion in an active form for interacting with TirM$^{EHEC}$, whether in solution or bound to a surface[17]. As sketched in Fig. 1, this design links the amino acid sequence of the CDRs of the Nb to short genomic segments of the *E. coli* carrier. This in turn facilitates application of ssDNA recombineering for targeted diversification of such DNA segments, eventually resulting in variability of the primary sequences of the Nb's CDRs.

**Generation of large repertoires of surface-displayed TD4 variants.** The experimental workflow followed to broaden the sequences of the CDRs of the surface-displayed anti-TirM$^{EHEC}$ Nb is sketched in Fig. 1 and described in the Methods Section. Basically, *E. coli* EcM1luxSAtir was first transformed with plasmid pORTMAGE3 (Supplementary Table S2), which upon thermal induction generates the Redβ ssDNA annealing protein (SSAP) for enabling penetration of synthetic oligos within the replication fork, along with a dominant negative allele of *mutL* (MutLE32K) that transiently supresses mismatch repair—thereby increasing occurrence in vivo recombineering events in at least four orders of magnitude[32–34]. The pORTMAGE3-bearing strain was then cyclically transformed with a cocktail of target-specific mutagenic 90-mer oligonucleotides. The whole of synthetic ssDNA used in the experiment included 3 distinct pools, each targeting one of the segments of the V$_{HH}$ gene that encoded domains CDR1, CDR2 and CDR3, respectively. These oligos were designed[34] to introduce 1.5 mutations on average per 100 nucleotides of the targeted sequence to thus generate an average of single amino acid changes within the 3 CDRs of the V$_{HH}$ protein. Following ten consecutive rounds of DIvERGE mutagenesis the DNA sequences corresponding to the V$_{HH}$ genes borne by *E. coli* were amplified from the merged genomic DNA extracted from the culture with primers TirDSF and TirDSR (Supplementary Table S3). The resulting DNA products where then submitted to Pacific Biosciences Circular Consensus (PacBio CCS) and Illumina amplicon sequencing for appraisal of the mutagenesis procedure, the outcome being summarized in Fig. 2 and Supplementary Fig. S1. The results show that after ten consecutive rounds of DIvERGE mutagenesis a large number of single-nucleotide replacements accumulated throughout the sequence corresponding to the domains CDR2 and CDR3 and (in way less pronounced fashion) in CDR1. While the lower diversification of this last segment could be traced to a suboptimal efficiency of DIvERGE at that target site (e.g. because of DNA secondary structures or reduced recombination), it could also happen that structural changes in that protein region are counterselected in vivo due to detrimental interference with the secretion system. Results of Fig. 2 show also that mutagenesis did not spread beyond the boundaries of the targeted CDR sequences and that in any case it was not above non-treated samples. Note that DIvERGE technology is documented to yield no detectable off-target changes even after 24 consecutive rounds of mutagenesis with pORTMAGE3[34] thereby ruling out unspecific effects in the workflow. In sum, these experiments benchmarked the procedure and warranted the search of new binding specificities in the mutated TD4 Nb as explained next.

**Selection of high-affinity anti-TirM$^{EPEC}$ Nbs by magnetic cell sorting (MACS).** In order to enrich the library of V$_{HH}$ variants derived from the TD4 Nb for those which have acquired the ability to bind the new antigen, the target TirM$^{EPEC}$ domain was purified (Supplementary Fig. S2) and labelled with biotin as described in the Methods Section. The *E. coli* cells bearing the

mutagenized V$_{HH}$ pool were next incubated with 250 nM the biotinylated antigen (Fig. 3a), washed to remove unbound Tir-M$^{EPEC}$ and then mixed with paramagnetic microbeads coated with streptavidin for MACS. The sample was subsequently passed through a MACS iron column held on a magnet. Cells coated with bound biotinylated antigen and anti-biotin microbeads were retained whereas those with no bound antigen on their surface were washed out the column and discarded. Finally, bacteria bound to the column were eluted, regrown and used to start another selection cycle (Fig. 3a). The same workflow was applied to equivalent *E. coli* cells displaying an unrelated anti-GFP Nb (negative control) and to the parental cells incubated with the original antigen TirM$^{EHEC}$ (positive control). Colony forming units (CFU) stemming from bacteria in the washout eluate (unbound fraction) and in the antigen-bound fractions were determined before and after MACS cycles for all samples (Supplementary Fig. S3a). The CFU figures show that before sorting, virtually all cells of the input library were in the unbound fraction. In contrast, after MACS, the percentage of bound cells was >20% (Supplementary Fig. S3a). Expectedly, cells displaying the control anti-GFP Nb were not retained in the column above background levels using biotinylated TirM$^{EPEC}$ antigen. Non-mutated *E. coli* cells displaying the parental TD4 Nb and treated with the original biotinylated TirM$^{EHEC}$ antigen were efficiently retained in the column (>90%) (Supplementary Fig. S3a). In order to monitor the emergence and evolution of the new binding activity of the progressively enriched library, aliquots of cultures resulting from each MACS cycle were incubated with 250 nM of biotinylated TirM$^{EPEC}$ or biotinylated BSA as control antigen), washed and stained with streptavidin-phycoerythrin (PE; see Methods). Samples were then inspected in a fluorescence flow cytometer along with controls of the non-diversified strain (*E. coli* EcM1*lux*SAtir) and the input library (diversified but unselected) treated under identical conditions. The results shown in Fig. 3b indicated that from the second cycle of MACS onwards, the selected populations were mainly composed of cells effectively and specifically binding TirM$^{EPEC}$. Additional rounds of selection enhanced the TirM$^{EPEC}$ binding signal on bacterial cells ca. 3-fold, which remained constant or showing a small reduction in the fourth and fifth cycles. Taken together, these data settled the efficacy of the workflow sketched in Fig. 3a for selecting V$_{HH}$ variants with potentially new specificities.

**Analysis of mutations in the evolved V$_{HH}$ clones.** In order to examine the changes entered during the DIvERGE procedure in the sequence of the anti-TirM$^{EHEC}$ Nb enabling binding to Tir-M$^{EPEC}$, the genomic regions of the V$_{HH}$ were amplified as above in each of the MACS cycles. The distribution of amino acid replacements was then determined through DNA sequencing of the resulting PCR products with PacBio CCS technology (see Methods). Figure 3c summarizes the emergence and distribution of the most frequent changes found in the clone library at each enrichment cycle. For examining individual clones, samples from the last MACS selection cycle were plated to obtain single colonies. Their V$_{HH}$ gene variants were subsequently amplified by PCR and separately sequenced. By matching the resulting discrete sequences with those of the pooled data we could calculate the occurrence and evolution of each variant relative to the overall mutant frequency after each round of MACS sorting (Fig. 3c and Supplementary Fig. S3c). Interestingly, some early mutant clones disappeared along the enrichment cycles while others stayed at relatively high frequencies compared to the rest. This analysis also revealed an unexpected high proportion of wild-type VHH sequences in the bound population after MACS selection. Although the basis of this phenomenon is not fully understood, it

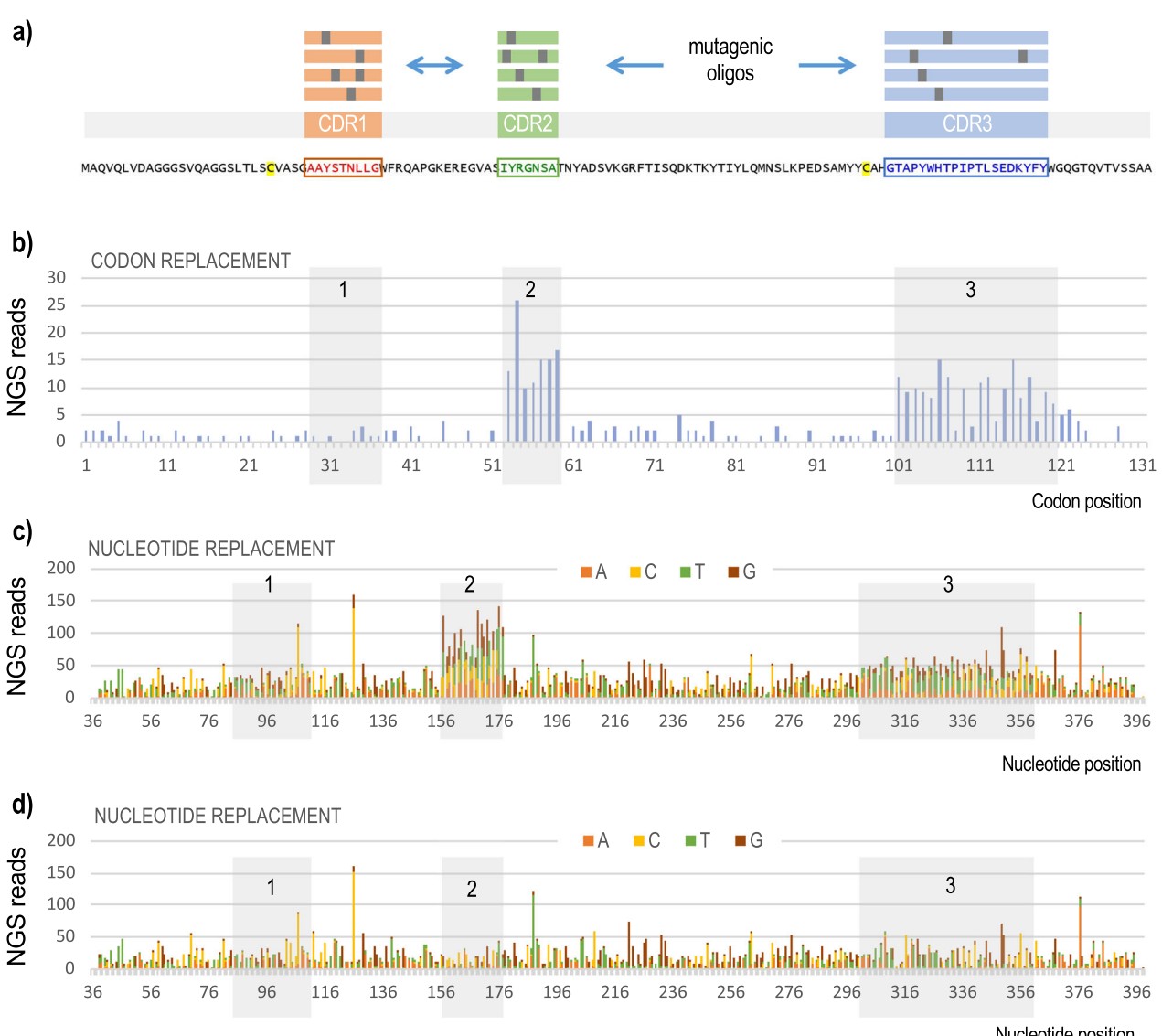

**Fig. 2 Distribution of nucleotide changes through the anti-TirM$^{EHEC}$ VHH sequence following mutagenic recombineering. a** DIvERGE is based on the use of mutagenic oligonucleotide pools that integrate in the three genomic loci encoding the complementarity determining regions (CDRs) of the VHH at stake and thus bring about generation of Nb libraries. The CDRs are colour-coded in the amino acid sequence of the parental Nb TD4. Two cysteines involved in the formation of disulfide bridges by post-translational modification are highlighted in yellow. **b** Distribution of codon replacements in VHH sequence as exposed by the results of PacBio sequencing. **c** Distribution of single nucleotide replacements after in vivo mutagenesis, as determined by sequencing with Illumina technology. The segments corresponding to CDR1, 2 and 3 are highlighted. **d** Negative control following the same procedure (NGS error) without mutagenic oligonucleotides. The lower mutagenic efficiency of the oligonucleotide pool targeting CDR1 may have resulted from a higher tendency to form secondary structures which could hinder the transformation process or the incorporation of these oligonucleotides during recombineering. The presence of replacements all over the VHH sequence, at low frequency, are likely to originate in the high error rate of Illumina sequencing (typically in the range of 0.1–0.2%[18, 46], as it appears both in the mutagenized libraries and the control experiments. An additional analysis of mutation distribution with PacBio long-read sequencing is shown in Supplementary Fig. S1.

may be caused by low-affinity interactions of wild-type Nb to TirM$^{EPEC}$ due to avidity effects favoured by the multivalent expression of the Nb on the bacterial surface and their stabilization with anti-biotin Ab (or Streptavidin) and magnetic beads[35,36]. Hence, clones with wild type V$_{HH}$ were excluded from further analysis. Eventually, mutant clones were identified that were gradually enriched along the MACS cycles, the most abundant being those with V$_{HH}$ encoding Nbs with single mutations in residues H107Y, T108R and D116G in their primary sequence (Fig. 3c; Supplementary Table S4). Amino acid replacements in these three mutants occurred in their CDR3 segment, which is coherent with earlier observations that largely trace specificity changes to modifications of CDR3 in Nbs[7]. The three

mutants were then subsequently picked for further analyses as described below. Their primary amino acid sequences are shown in Supplementary Table S4.

**Binding of TirM antigens by the evolved nanobodies displayed on the bacterial surface.** *E. coli* strains displaying on their surface nanobodies TD4$^{H107Y}$, TD4$^{T108R}$ and TD4$^{D116G}$ (see above) were tested for their ability to bind the original antigen TirM$^{EHEC}$ in comparison with the one used for selection (TirM$^{EPEC}$). For this, cells from cultures expressing the intimin-VHH fusions were incubated with either 250 nM of biotinylated TirM$^{EPEC}$ or 50 nM biotinylated TirM$^{EHEC}$, followed by staining with streptavidin-

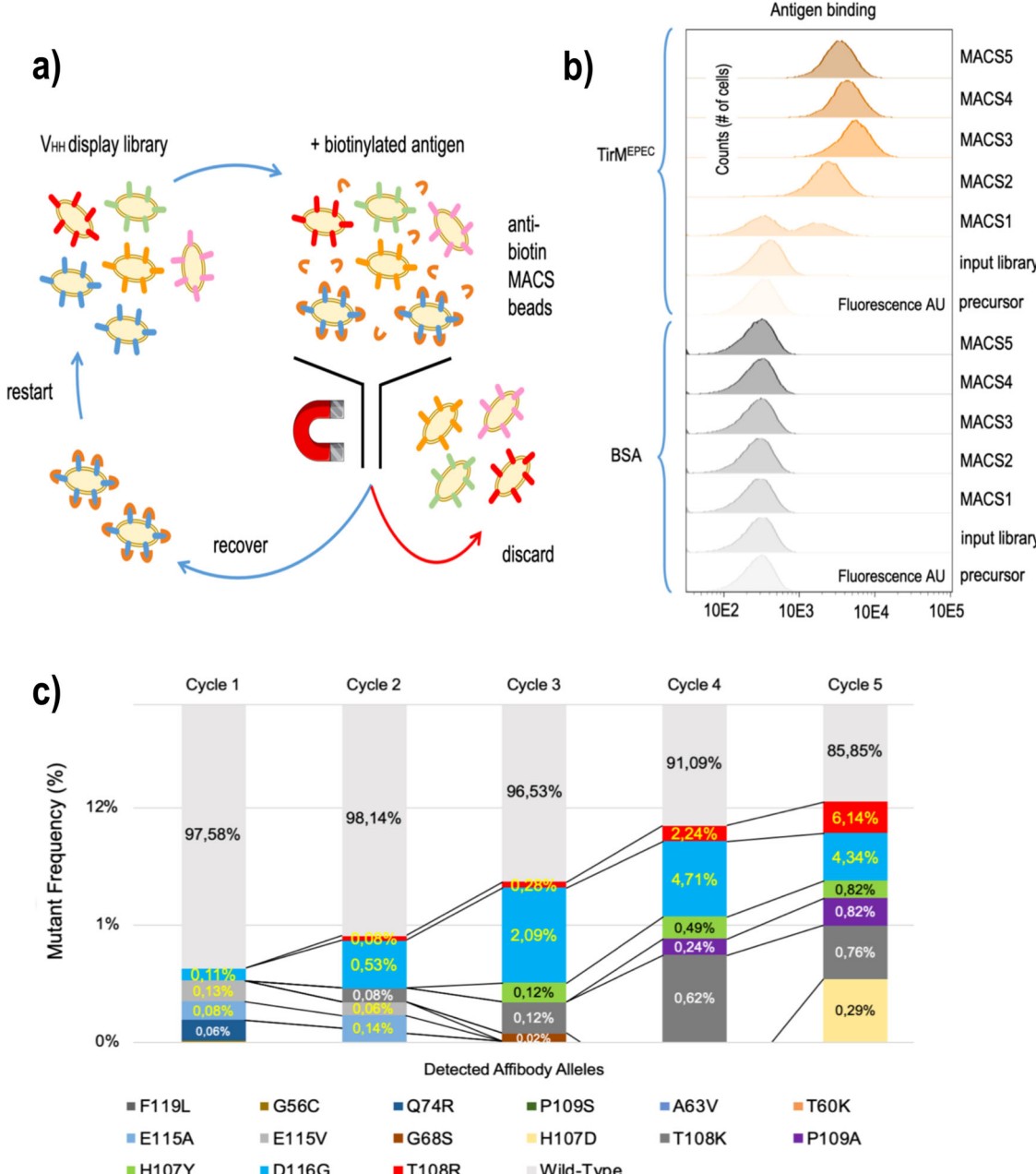

**Fig. 3 Pipeline for enrichment of new binding specificities in evolved nanobodies. a** Libraries of *E. coli* bacteria displaying the VHH variants resulting from 10 DIvERGE cycles are mixed with a biotinylated antigen and then with anti-biotin magnetic-activated cell sorting (MACS) beads. Only the cells displaying a Nb that specifically binds the antigen can subsequently be bound to the MACS column. Wash and subsequent elution results in the enrichment of the libraries with antigen binding clones. **b** Evolution of new binding capabilities in *E. coli* populations expressing anti-TirM nanobodies following expression and display of the VHHs. Bacteria retrieved from each of the MACS cycles were exposed to either biotinylated BSA or with biotinylated TirM^EPEC, followed by fluorescent staining with streptavidin-phycoerythrin. The results show how the bacterial pool is enriched with clones that bind the new antigen TirM^EPEC right after the first round of immunomagnetic selection. After the second cycle and beyond, the population seems predominantly composed by clones able to bind the new antigen. Note that these analyses just reflect bulk population behaviour, but they do not say anything yet on the nature of the mutations. **c** Progress of VHH population composition along cycles of immunomagnetic enrichment of TirM^EPEC-binding bacteria. The frequencies and points of emergence of fifteen representative Nb mutants displayed by bacteria captured in subsequent MACS cycles are shown in logarithmic scale in respect to those bearing the wild-type VHH specimen (see Supplementary Fig. S3c for a more detailed evolution of variants).

allophycocyanin (APC; see Methods). Bacteria were then subjected to fluorescence flow cytometry analysis and the results are shown in Fig. 4. Inspection of the histograms revealed that while control cells displaying the non-evolved TD4 Nb expectedly had no noticeable interaction with TirM^EPEC, the counterparts expressing TD4^H107Y, TD4^T108R and TD4^D116G exhibited a

considerable binding to the new antigen (Fig. 4a). In addition, the evolved nanobodies maintained strong binding signals for the original target TirM^EHEC, similar to the parental Nb (Fig. 4c). These results suggested that the evolved nanobodies had expanded their binding capacity towards the novel antigen—rather than switching specificities completely. This outcome was welcome, as

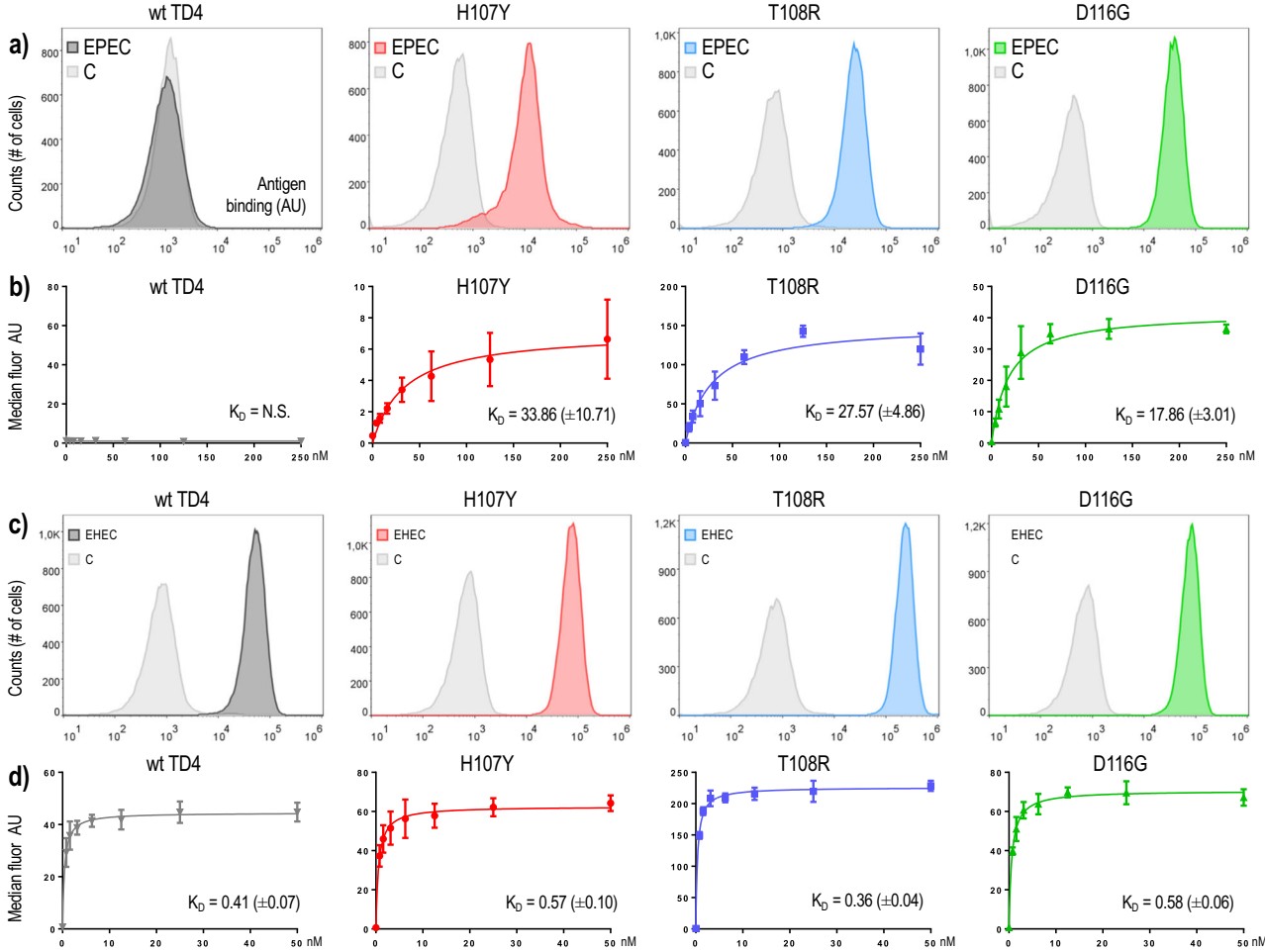

**Fig. 4 Binding properties of the nanobodies displayed on the bacterial surface.** Bacterial clones displaying the Nb variants indicated in each case on the cell surface were incubated with biotinylated antigens and streptavidin-allophycocyanin and then analyzed by fluorescence flow cytometry. TD4 corresponds to the parental Nb which was the template for the generation of libraries. H107Y, T108R and D116G are the names of three representative mutants and refer to the replacement they carry in the amino acid sequence of the Nb. Panels **a** and **b** show the interaction of these nanobodies with the new antigen TirM$^{EPEC}$. Note that in contrast to the parental Nb TD4, the replacements in the amino acid sequence of the new variants lead to a gain of function that enables them to recognize the new target. Panels **c** and **d** show the interaction with the former antigen TirM$^{EHEC}$. The four nanobodies tested still bind the former antigen TirM$^{EHEC}$. Control samples—tagged C in the cytometry panels **a** and **c**—were incubated only with buffer and streptavidin-allophycocyanin. Antigen binding was based on fluorescence intensity values measured in arbitrary units (AU). For each concentration of antigen in **b** and **d**, median fluorescence values of four independent replicas were considered. Estimated apparent $K_D$ ( ± standard error) values are indicated. N.S not significant. Note different scales in x axes.

eventually we pursued Nbs able to bind simultaneously to different types of TirM variants.

To gain some quantitative insight into the expansion in antigen specificity, we next measured the apparent affinities of the TD4 variants toward EHEC and EPEC TirM antigens when displayed on the surface of *E. coli*. For this, bacteria were incubated with different concentrations of the biotinylated antigens ranging from 0 to 250 nM in the case of TirM$^{EPEC}$ (Fig. 4b) and 0–50 nM for TirM$^{EHEC}$ (Fig. 4d). The median fluorescence intensity values were fitted by a non-linear regression curve based on the model $Y = B_{max} \cdot X \cdot (K_d + X)^{-1}$, where Y is the fluorescence intensity, Bmax is the fluorescence signal extrapolated to very high concentrations of antigen, X is the concentration of the antigen and $K_d$ is the equilibrium dissociation constant (i.e. the antigen concentration that binds to half of the binding sites at equilibrium). Based on best fit estimations, the apparent equilibrium dissociation constant ($K_d$) for the new antigen TirM$^{EPEC}$ exposed apparent binding affinities of H107Y, T108R and D116G mutants in a similar range, between 17 to 33 nM

(Table 1). The $K_d$ value for the parental TD4 for TirM$^{EPEC}$ could not be determined as there was no significant binding to at any of the concentrations tested. Interestingly, parental and evolved nanobodies showed similar apparent $K_d$ for the former antigen TirM$^{EHEC}$, which were in all cases in the subnanomolar range (c.a. 0.5 nM; Table 1). Therefore, the evolved TD4 variants still maintain an apparent higher affinity for the original antigen TirM$^{EHEC}$ than for the new antigen TirM$^{EPEC}$. These data suggested that the effect of the mutations create new interactions with TirM$^{EPEC}$ while maintaining strong interactions with the former antigen. In sum, while these test with whole cells may not be rigorously quantitative, the resulting apparent $K_d$ values (Table 1) were indicative of a strong binding of the evolved Nbs to EPEC TirM and we set out to investigate the intrinsic properties of the evolved Nbs as purified proteins.

**Binding affinities of the evolved monovalent nanobodies against TirM antigens.** Although the experiments above exposed important features of the Nb variants TD4$^{T108R}$, TD4$^{H107Y}$, and

**Table 1 Apparent $K_D$ values of evolved nanobodies (x$10^{-9}$ M) by flow cytometry of *E. coli* bacteria.**

| Nb mutant | Nanobody experiment by flow cytometry (multivalent) | | Nanobody experiment by ELISA$^\alpha$ (monovalent) | |
| --- | --- | --- | --- | --- |
| | New antigen (TirM$^{EPEC}$)[a] | Former antigen (TirM$^{EHEC}$)[a] | New antigen (TirM$^{EPEC}$)[a] | Former antigen (TirM$^{EHEC}$)[a] |
| Nb TD4 | N.S. | 0.41 (±0.07) | N.S. | 5.55 (±1.19) |
| | N.S. | [0.26 to 0.55] | N.S. | [3.02 to 8.08] |
| Nb H107Y | 33.86 (±10.71) | 0.57 (±0.10) | 20.08 (±2.31) | 47.74 (±12.38) |
| | [11.99 to 55.73] | [0.36 to 0.78] | [15.19 to 24.97] | [21.49 to 73.98] |
| Nb T108R | 27.57 (±4.86) | 0.36 (±0.04) | 176.3 (±47.11) | 4.61 (±9.91) |
| | [17.64 to 37.50] | [0.29 − 0.43] | [76.41 to 276.2] | [Very wide] |
| Nb D116G | 17.86 (±3.01) | 0.58 (±0.06) | 397 (±140.9) | 136.6 (±105.8) |
| | [11.71 to 24.01] | [0.45 to 0.71] | [98.25 to 695.7] | [87.70 to 360.9] |

[a] Best-fit values (standard error) and [95% confidence intervals] are shown for each of the nanobodies in both experiments.
$^\alpha$ ELISA: enzyme-linked immunosorbent assay.
$K_d$ values and confidence intervals were estimated using the function "One site - Specific binding" from the software GraphPad (Prism).

TD4$^{D116G}$, the data on their apparent affinities might be influenced by their expression in multimeric form on the surface of bacteria[8]. To clarify this, the mutant proteins were purified as monovalent polypeptides following their secretion by *E. coli*. To this end, the corresponding $V_{HH}$ sequences were cloned in fusion with the C-terminal secretion signal of hemolysin A (HlyA) and the resulting plasmid constructs were then placed in a dedicated *E. coli* strain that expresses HlyB and HlyD components of the hemolysin protein secretion system (see Methods Section[22]). The strains bearing these fusions were grown in LB, induced with IPTG, and the secreted nanobodies purified from the culture media (Supplementary Fig. S4). Controls included one strain secreting the non-evolved parental TD4 and another strain producing a non-related Nb (Vamy[22]). Affinities were determined by coating ELISA plates with purified His-tagged versions of Tir-M$^{EHEC}$ and TirM$^{EPEC}$ and the values estimated as shown in Fig. 5. In this case, results clearly indicated that the former antigen TirM$^{EHEC}$ is best recognized by the non-evolved TD4 ($K_d$ = 5.5 ± 1.19 nM) in a fashion about one order of magnitude better than TD4$^{H107Y}$ ($K_d$ = 47.74 ± 12.38 nM) and >20-fold in respect to TD4$^{T108R}$ and TD4$^{D116G}$. The new antigen TirM$^{EPEC}$ was not recognized by the parental TD4 Nb at any concentration tested but, the evolved counterparts manifest their newly gained interaction to the new target with apparent $K_d$ values of 20.8 ± 2.31 nM (TD4$^{H107Y}$), 176.3 ± 47.11 nM (TD4$^{T108R}$) and 397 ± 140.9 nM (TD4$^{D116G}$), respectively. Divergence of the apparent $K_d$ values determined with the monovalent Nbs with those estimated with *E. coli* bacteria are likely reflecting avidity effects due to the distinct physics of the Nb-antigen interaction in monovalent (purified) versus multivalent (cell-displayed) formats[35,36]. In this respect, purified TirM$^{EPEC}$ showed a dimeric behaviour in solution as determined by size exclusion chromatography (Supplementary Fig. S2d), which may cause the binding avidity increase observed with multivalent Nb on the bacterial surface. Taken together, these results demonstrate the ability of the hereby described workflow to expand the molecular target of a given Nb towards a different one.

**Structural predictions of target expansion by evolved nanobodies.** For the sake of gaining insight into the changes that endowed TD4$^{H107Y}$, TD4$^{T108R}$ and TD4$^{D116G}$ with new properties, we threaded the primary amino acid sequence of all the TirM variants in the well-conserved tridimensional structure of known nanobodies determined by X-ray crystallography using bioinformatic tools (Methods Section). The resulting shape of the original TD4 model was then overlapped pair-wise with each of the other 3 evolved variants, with the outcome shown in Fig. 6. The root mean square deviation (RMSD) values between each

pair of structures (indicative of divergence from each other) were very low, indicating structures to be virtually identical excepting for the very sites where mutations occurred. Inspection of the amino acid changes revealed that in position 107 a basic histidine is replaced with a bulkier, uncharged and protruding polar tyrosine. In the nearby 108 position an uncharged and polar threonine residue was exchanged by a basic and also bulkier/protruding arginine. This exposed the importance of the site location around 107–108 for recognition of the new antigen, perhaps by direct contacts with specific residues of the TirM$^{EPEC}$ structure that were not amenable to binding by the non-evolved TD4 Nb[21]. In contrast, an acidic aspartate at position 116 was replaced with a neutral and smaller glycine, suggesting that in this case the change likely eliminated an inhibitory repulsion between a charged amino acid and an incompatible site of TirM$^{EPEC}$. These replacements are also predicted to change the orientation of the new amino acid in respect to the original one, although due to the size of the side chains, this was more pronounced in the case of H107Y and T108R. In any case, replacement of a single amino acid in TD4 was sufficient gain of a new binding specificity while their former capacity to bind TirM$^{EHEC}$ was not abolished. This was noticed in the three purified nanobodies, but the tradeoffs were different in each case. TD4$^{H107Y}$ was the one with higher TirM$^{EPEC}$-binding affinity and the mildest loss affinity for TirM$^{EHEC}$. In contrast, acquiring a new binding towards Tir-M$^{EPEC}$ for TD4$^{T108R}$ and TD4$^{D116G}$ was at a higher cost for their interaction with the former antigen.

**Conclusion.** The results above showcase the power of merging surface display of Nbs with genomic site-specific diversification elicited by DIvERGE technology in combination with MACS-based cyclic selection. The CDRs of the encoded Nb can be easily targeted by ssDNA mutagenic recombineering[18] of the $V_{HH}$ gene integrated as single copy in the bacterial chromosome and fused to intimin outer membrane anchor domain for bacterial display, and the resulting library directly screened and characterized for binding to a new antigen target of choice. However, the apparent affinities for the evolved Nbs on the bacterial surface do not exactly match with their binding affinities observed as purified monovalent proteins, likely reflecting complex avidity effects of the multimeric format of the Nb displayed on the bacterial surface and the dimeric nature of the purified antigen (TirM$^{EPEC}$). Nonetheless, the hereby proposed experimental pipeline delivers a suite of Nbs endowed with a range of affinities for the target that cover different needs. In the specific instance addressed in this work we pursued to expand the therapeutic potential of a specific anti-TirM$^{EHEC}$ Nb that inhibits the attachment of EHEC bacteria to human intestinal tissue towards EPEC strains, which are a

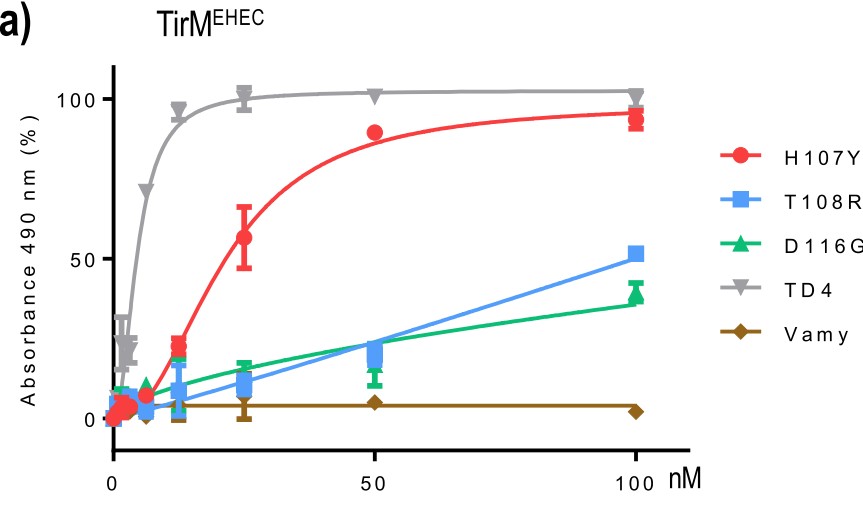

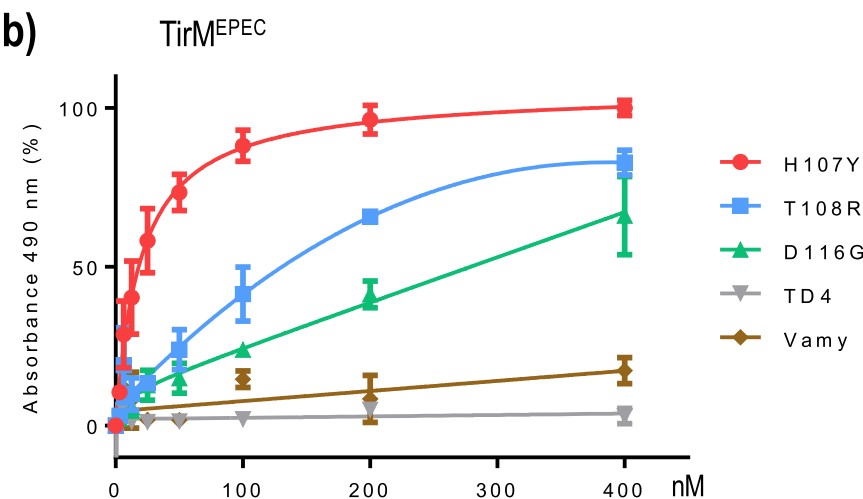

**Fig. 5 ELISA of purified nanobodies against TirM antigens.** Interaction of the evolved nanobodies with the former antigen TirM$^{EHEC}$ (**a**) and new antigen TirM$^{EPEC}$ (**b**). Vamy is a Nb specific for alpha amylase used as an unspecific control. Note that TD4 does not bind the new antigen, H107Y binds quite well the former and new antigen (it may be considered as a generalist) while T108R and D116G have experienced a minor loss of their ability to bind the former antigen while gaining the ability to bind the new antigen (note that the x axis scale is different in each graph).

major cause of acute and chronic diarrhoea in infants[23]. But one can envision a number of scenarios which make the workflow of Figs. 2 and 3 exceptionally worthy also e.g. for quickly developing existing VHHs against viral proteins towards emerging variants for diagnostic or therapeutic purposes[37–39]. This can be specially useful when thereby selected Nbs are subsequently bulk-produced in bacteria with the hemolysin export system—also adopted in our experimental roadmap above. While alternative microbial platforms for in vivo generation of Nb repertoires do exist to the same end[10,12,37,40,41] we believe that the presented pipeline is simple and contributes to ongoing efforts towards point-of-care technologies for diagnostics and other medical and environmental needs[24,42–44].

## Methods

**Materials, oligonucleotide synthesis and general procedures.** *E. coli* strains used in this work can be found in Supplementary Table S1. Plasmids and other genetic constructs are listed in Supplementary Table S2. Recombinant DNA manipulations, media preparation and culture conditions followed standard protocols[45]. Purification of TirM$^{EHEC}$ and TirM$^{EPEC}$ was done as described in

refs. [21,22]. Protein concentrations were estimated using Protein assay reagent (BioRad) or BCA protein assay (Thermo Fisher Scientific). Whenever required, proteins were analyzed in denaturing polyacrylamide gels.

**DIvERGE oligonucleotides.** ssDNAs for recombineering and library diversification were designed to minimize secondary structure, to target the lagging strand of the DNA replication fork and to incorporate 1-2 mutations per CDR1, CDR2 and CDR3 segments of the V$_{HH}$ sequence. For this, 3 randomized regions of the mutagenic oligonucleotides (Supplementary Table S3) were synthesized with 0.5%–0.5%–0.5% soft-randomization with the three other possible nucleobases other than the wild type. This creates an average chance of having a SNP at a given, randomized position of 1.5% (i.e. 1.5 mutation on average per 100 target bases), which translates into a mutation rate of 1–2 SNP per gene in the resulting library[33,34]. Each mutagenic oligo pool was synthesized on an ABI3900 DNA synthesizer (Applied Biosystems) following a modified protocol based on phosphoramidite chemistry. The synthesis cycles were applied on a controlled pore glass as solid support and were carried out as follows. First, deprotection was done with trichloroacetic acid 3% (w/v) in dichloromethane. Second, incoming phosphoramidite was dissolved in $55 \times 10^{-3}$ M anhydrous acetonitrile, then mixed with the other three amidites in and subsequently coupled by activation with 5-ethylthio-1H-tetrazole. Third, capping was done with acetic anhydride 10% (v/v) in anhydrous tetrahydrofuran and N-methyl-imidazole 16% (v/v) and pyridine 10% (v/v) in an anhydrous tetrahydrofuran solution. Finally, the oxidation was done with a

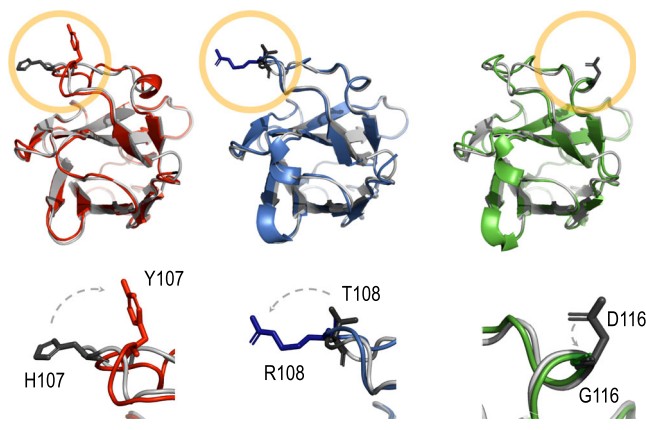

**Fig. 6 Comparison of the structural models of the Nb variants.** The upper row shows the different configurations of each new Nb compared to the structural model of the precursor protein. H107Y (red), Nb T108R (blue) and Nb D116G (green) are shown aligned to TD4 (grey). The bottom row shows a blowup of the replacement positions with former residues in dark grey and the new ones in dark red, dark blue and dark green, respectively. Changes involve replacement of a basic residue by a bulkier, uncharged and polar counterpart in the case of H107Y, the change an uncharged and polar amino acid by a basic and larger one in T108R and the substitution of an acid residue by a neutral and smaller one in D116G. Although more evident in H107Y and T108R, a change in the orientation of the new amino acid with respect to the original alignment occurs in all the three replacements. The RMSD values for the structural alignments with TD4 were all <1 (i.e. 0.781810 for H107Y, 0.487231 for T108R and 0.637916 for D116G) which is indicative that the overall structures of all these nanobodies can be considered almost identical.

solution of 5 g of iodine per litre of a mixture of pyridine:water:tetrahydrofuran in the ratios 0.5:2:97.5, respectively. Cycles were repeated until the 90th nucleotide and then, the DNA strands were cleaved from the solid support with concentrated ammonia. Crude oligos were purified by high-performance liquid chromatography (HPLC) on a Shimadzu ADVP-10 HPLC system. HPLC fractions were concentrated and dimethoxytrityl (5′-DMTr) protecting groups were removed using a PolyPak column (Glen Research) following the manufacturer's protocol. Oligonucleotides (Integrated DNA Technologies) were eluted, lyophilized and suspended in TE buffer (1X) pH 8.0. Their sequences are listed in Supplementary Table S3.

**Genomic DNA sequence diversification with DIvERGE**. A fresh single colony of *E.coli* EcM1 bearing pORTMAGE3 (Addgene Plasmid ID #72678) taken from LB$_{a,Km}$ was inoculated into 2 mL LB$_{Km}$ and incubated overnight at 30 °C, 170 rpm. This cell culture was used to prepare a 1:100 dilution in 10 mL LB$_{Km}$ that was then incubated at 30 °C and 170 rpm until reaching 0.4 OD$_{600nm}$. Next, this was incubated 15 min in a water bath at 42 °C with shaking at 250 rpm and then the flask containing the cells was kept 10 min in ice. The grown cells were harvested by centrifugation at 4000 rpm for 8 min at 4 °C and supernatant discarded. Sedimented cells were washed by resuspension in 10 mL ice-cold water, centrifugation at 4000 rpm for 8 min at 4 °C and supernatant discarded. The sediment of cells from the last wash was suspended in 160 μL of ice cold dH$_2$O and kept on ice. A mix of 3 μL with primers pools tirM1, tirM2 and tirM3 (Supplementary Table S3) in 1:1:1 ratio were incorporated to 40 μL of competent cells in a 1.5 mL tube and the resulting mix was kept on ice for 5 min. These cells were electroporated, then suspended in 6 mL of TB and subsequently kept at 30 °C with shaking at 170 rpm. After 60 min, 10 mL LB supplemented with kanamycin were added and the mix was kept at 30 °C in a shaker until 0.4 OD$_{600nm}$. Then, 10 mL of this culture was taken to prepare electrocompetent cells in order to start a new DIvERGE cycle. The other 5 mL was grown overnight, then OD$_{600nm}$ was measured and 1.5 mL was used for genomic DNA extraction and another 1.5 mL to prepare a stock tube in glycerol 20% to store at −80 °C.

**Biotinylation**. Fractions of the antigens EHEC TirM or EPEC TirM were diluted in 1.5 mL PBS to 1 mg ml$^{-1}$. The NHS-Biotin (Sigma-Aldrich B2643) was dissolved in DMSO and immediately added to these protein solutions in a biotin:protein molar ratio of 20:1 and the mix was incubated for 30 min at RT with gentle agitation. Then 1 M Tris-HCl, pH7.5 was added for a final concentration of 50 mM and the mixture was incubated for 1 h on ice. The sample was concentrated to a final volume of 500 μL in a 3 kDa Centricon (Merck) by centrifugation at 5000 g.

Finally, the efficiency of biotinylation was checked in a Western blot revealed with POD-anti-biotin (horseradish peroxidase conjugated with Streptavidin, Roche).

**Magnetic-activated cell sorting (MACS)**. An amount of 10^8 bacteria were washed by centrifugation (5 min, 4000 g, 4 °C) and supernatant discarded and then were suspended in 2 mL of BSA 0.5% in PBS 1X. The same process was repeated again from the beginning. Then, the cells were washed again and suspended in 200 μL of 250 nM of the biotinylated antigen TirM$^{EPEC}$ in a solution of BSA 0.5% in PBS 1X, mixed gently and incubated 1 h at 4 °C. These cells were washed three times, suspended in 80 μL of a solution of BSA 0.5% in PBS 1X, mixed gently with 20 μL of anti-biotin microbeads and incubated for 30 min at 4 °C. Then, the cells were washed three times and suspended in 500 μL of BSA 0.5% in PBS 1X. A column was placed on the magnet and rinsed with 1.5 mL of PBS + BSA 0.5%. Then, the cell suspension was applied onto the column and the flow-through collected. The column was washed three times with 500 μL of PBS + BSA 0.5%, and the three flow through were collected together in the same tube. The column was removed from the separator and placed on a suitable collection tube. Next, 2 mL of PBS + BSA 0.5% were added onto the column. The magnetically labelled cells were immediately flushed out by firmly pushing the plunger into the column.

**Flow cytometry monitoring of binding capabilities in the libraries after each MACS cycle**. Cells were grown overnight in 20 mL cultures at 30 °C and 170 rpm. A volume of cells corresponding to 0.25 OD$_{600nm}$ was centrifuged at 4000 g for 2 min at 4 °C followed by 20,000 g for 2 min at 4 °C. The supernatant was discarded and the sediment was washed three times with 1 ml of pre-chilled and filter-sterilized PBS and centrifugation at 4 °C and 4000 g for 2 min followed by 20,000 g for 2 min at 4 °C and supernatant discarding. Then, cells were harvested by centrifugation (at 4 °C at 4000 g for 2 min followed by 20,000 g for 2 min at 4 °C) and the sediment suspended in 500 μL of ether, 250 nM of biotinylated TirM$^{EPEC}$ in PBS or 250 nM of biotinylated BSA in PBS. Cells were incubated for 90 min at 4 °C, washed three times with PBS as before and labelled with streptavidin-phycoerythrin. The cells were washed with PBS and fluorescence was quantified in a flow cytometer (Gallios, Bekman Coulter).

**In vivo determination of Nb-antigen affinity by flow cytometry**. The strains EcM1T108R, EcM1H107Y and EcM1D116G were inoculated in separate flasks with 25 mL of LB$_{km}$ and EcM1LuxSATir in a flask with 25 mL of LB. These four flasks were subsequently incubated overnight at 30 °C and 170 rpm. From each culture, a volume corresponding to 1 OD$_{600nm}$ was centrifuged at 4000 g for 2 min at 4 °C followed by 20,000 g for 2 min at 4 °C, the supernatant was discarded and the sediment was washed three times with 1 ml of pre-chilled and filter sterilized PBS and centrifugation at 4 °C and 4000 g for 2 min followed by 20,000 g for 2 min at 4 °C. Then, the cells were harvested by centrifugation (at 4 °C at 4000 g for 2 min followed by 20,000 g for 2 min at 4 °C) and the sediment suspended in 500 μL of PBS. Next, 9 fractions with 25 μL of this cell suspension were centrifuged at 20,000 g and 4 °C for 2 min, the supernatants discarded and the cells incubated 90 min at 4 °C with a fixed amount of biotinylated antigen and increasing volumes of PBS from 0.25 to 1.6 mL to attain a final concentration range between 0 nM and 250 nM TirM$^{EPEC}$ or 0 nM and 50 nM TirM$^{EHEC}$, as indicated. After incubation, cells were centrifuged at 4000 g and 4 °C for 2 min followed by 20,000 g at 4 °C for 2 min, then washed twice with 1 mL of filtered and sterilized PBS at 4 °C and labelled with Streptavidin-allophycocyanin. After a final washing step with PBS, the median fluorescence intensity of APC was quantified in a cytometer (Gallios, Beckman Coulter). Data of median fluorescence intensity relative to maximum median fluorescence intensity obtained from the cytometer were plotted against the concentration of biotinylated antigen to estimate the dissociation constant ($K_D$). Curve was fitted according to non-linear least squares regression method and one site-specific binding saturation kinetics model using the data analysis tool in Prism software (GraphPad).

**Isolation and sequencing of genomic DNA**. Bacterial DNA was isolated from fresh overnight cultures of the libraries rMAGE1, rMAGE2, rMAGE3, rMAGE4, rMAGE5 and rMAGE5 control with the kit UltraClean® Microbial DNA Isolation Kit (Mo Bio, QIAGEN). Fractions of the isolated genomes were used as templates for PCR with the primers tirDSF and tirDSR and the obtained amplicons were purified and sent for sequencing with Illumina MiSeq set-up. Isolated genomic DNA from the libraries rMAGE5, rMAGE10, MACS1EHEC, MACS1CR, MAC-S1EPEC, MACS2EPEC, MACS3EPEC, MACS4EPEC and MACS5EPEC was used as template for a PCR with the primers tirDSF and tirDSR and the resulting amplicons were cleaned NucleoSpin® Gel and PCR Clean-up (Macherey-Nagel) and sent for PacBio circular consensus sequencing (CCS). For screening and for sequence confirmation, colony PCR products of around 1060 bp were obtained with the oligonucleotides CheckSATir_F and CheckSATir_Rev using Q5DNApolymerase (New England Biolabs) and the program as follows: 30 s at 98 °C, 30 cycles (15 s at 98 °C, 45 s at 70 °C and 90 s at 72 °C) and for final extension 120 s at 72 °C. PCR products were cleaned with the kit NucleoSpin® Gel and PCR Clean-up (Macherey-Nagel) and sent to Macrogen for sequencing with the oligonucleotide CDRI_Fwd. Results were used to estimate the frequency of mutations in the libraries.

**Plasmids for bulk production and purification of nanobodies**. Changes found in the DNA of the evolved VHH variants were first recreated in the frame of the wild-type TD4 sequence. To this end pCANTAB6_VhhTD4 plasmid (Supplementary Fig. S2) was used as DNA template for PCR using Q5 DNA polymerase (NEB) in three different reactions each containing the oligos 1 and 2 (for H107Y), 3 and 4 (for T108R) or 5 and 6 (for D116G). Conditions were 30 s at 98 °C, (15 s at 98 °C, 30 s at 65 °C, 3 min at 72 °C) x 18 cycles and 5 min at 72 °C. Mixes were subsequently digested 1 h at 37 °C with DpnI (NEB) and used for electroporation in *E. coli* CC118. Afterwards, each of the modified plasmids was isolated and confirmed by sequencing using the oligos 7 and 8 (Supplementary Table S3). The resulting plasmids were then named pCANTAB6_H107Y, pCANTAB6_T108R and pCANTAB6_D116G. In parallel, plasmid pEHLYA5_Vamy was isolated, digested with SfiI and NotI and the resulting DNA fragments were separated by electrophoresis in agarose gel. Then, the vector pEHLYA5 in the band corresponding to 3612 bp was cleaned with Nucleospin Gel and PCR clean-up kit (Macherey-Nagel). Plasmids pCANTAB6_H107Y, pCANTAB6_T108R and pCANTAB6_D116G were isolated, digested with SfiI and NotI and the DNA fragments separated by electrophoresis in agarose gel. Each DNA band of 400 bp containing a VHH gene variant was purified and ligated, to the SfiI/NotI-digested pEHLYA5 and the ligation mixed transformed in *E. coli* DH5α. Plasmids from clones growing in LB, ampicillin (100 µg/ml) and glucose (20%) were sequenced with the oligonucleotides 9 and 10 and, once verified were named pEHLYA_H107Y, pEHLYA_T108R and pEHLYA_D116G, respectively.

**Expression and purification of antigens and nanobodies**. Expression and purification of his-tagged TirM EPEC and EHEC antigens was conducted exactly as described previously[21]. Purification of the Nbs variants from culture supernatants using the hemolysin secretion system was done as explained before[22] with minor modifications. Briefly, *E. coli* HB2151 bacteria with pVDL9.3 (HlyBD transporters) and pEHLYA-derivatives with Vamy, TD4, and mutant Nbs fused to the C-terminal end of hemolysin A, were plated on LB agar with glucose and the proper antibiotics (Ampicillin and Chloramphenicol). One isolated colony of each strain was grown in aerobic conditions overnight at 30 °C in 25 mL of LB with glucose 2% and antibiotics. The resulting culture was used to prepare 200 mL of at 0.05 $OD_{600nm}$ in LB with antibiotics that was grown until 0.4 $OD_{600nm}$. Following that, IPTG to $10^{-3}$ M final concentration was added and the culture was further incubated for 6 h at 30 °C and 100 rpm. Next, the culture was centrifuged 4000 *g* for 15 min, the supernatant was passed to a new tube and centrifuged 4000 *g* for 15 min, the supernatant was passed again to a new tube and centrifuged 4000 *g* for 15 min, supernatant was collected in a new tube and supplemented with PMSF to $10^{-3}$ M final concentration. Each volume was supplemented with PBS 10x to a final PBS 1X concentration. Samples containing the thereby secreted [His]$_6$-tagged VHH-HlyA fusions were next passed through Talon CellThru resin (Clontech). For this, 2 ml of this metalloaffinity matrix, was placed in a 15 ml tube and it was subsequently washed by centrifugation 2 min at 700 *g*, supernatant discarding and suspension in 10 mL of PBS 1X. The resin was washed again, then it was centrifuged 2 min at 700 *g*, the supernatant was discarded and the resin was added to the suspension containing the secreted Nbs in 50 mL tubes. The mix was incubated in a spinning wheel and, after 30 min, it was added gradually to a 0.7 × 5 cm chromatography column (Bio-Rad). Once packed, this resin was washed in the column by passing gradually up to 30 mL of $5 × 10^{-3}$ M imidazole in PBS (1X) through the packed resin. Then, the protein was eluted in 1 mL fractions of 0.15 M imidazole in PBS (1X). Fractions were stored at 4 °C until needed.

**Enzyme-linked immuno assay**. ELISA conditions were based on a modified version of a previously described protocol[21,22]. Briefly, Maxisorp 96-well immunoplates (Nunc) were coated by adding 50 µL/well of a solution with the antigen EPEC TirM or EHEC TirM at 10 µg mL$^{-1}$ in PBS (1X) and incubating overnight at 4 °C. Then, blocking was done by discarding the liquid from the plate and adding 200 µL/well of PBS (1X) with skimmed milk (3%) and incubation for 60 min at 23 °C. Next, the liquid in the plate was discarded and the plate washed three times. Each wash consisted in adding 200 µL/well of a solution made of Tween 20 (0.05 %) in PBS (1X), incubation for 5 min at 23 °C and liquid discarding. Then, 50 µL with a Nb in a solution of skimmed milk (3%) in PBS (1X) was added per well in the range of 0–100 nM in the case of TD4 and 0 to 400 nM in the case of H107Y, T108R or H116G. Following incubation was 90 min at 23 °C. Next, the liquid was discarded and the plate washed three times, each consisting in adding 200 µL/well of Tween 20 (0.05 %) in PBS (1X), incubation for 5 min at 23 °C and liquid discarding. Mouse antibody anti-Etag diluted 1:2000 in a solution of skimmed milk (3%) in PBS (1X) was added and the plate was incubated for 60 min at 23 °C. Then, the plate was washed three times. Following that, the anti-mouse-POD antibody was diluted 1:1000 in the solution with skimmed milk (3%) in PBS (1X), the mix was added to the wells and the filled plate was incubated for 60 min at 23 °C. Then, the liquid was discarded and the plate was washed three times. Finally, results were developed as follows. One o-phenylenediamine tablet (OPD, Sigma) and 10 µL hydrogen peroxide (30 % w/v) were solved in 20 mL phosphate-citrate buffer (pH 5) and 80 µL/well was added from the mix. The plate was incubated for 20 min at 23 °C, protected from light and then, the reaction was stopped with 20 µL HCl (3 M)/well. Finally, $OD_{490nm}$ was determined using a microplate reader (iMark ELISA plate reader, Bio-Rad).

**Structural alignment of nanobodies**. The amino acid sequences of the Nbs (table IV) were introduced as input in FASTA format in i-TASSER online server (https://zhanglab.ccmb.med.umich.edu/I-TASSER/) to obtain the output which included each structural model file. The models of the structural alignments with the parental Nb TD4 were obtained with the function CEalign in PyMOL software. Alignment images were edited in PyMOL and exported as files with the extension png. Next, image cuts to show specific details were done with the software Fiji (https://imagej.net/Fiji).

**Statistics and reproducibility**. In Fig. 4b, d, for each of the concentrations of antigen, median fluorescent intensity values of four independent experimental replicas measured by a cytometer (Gallios, Beckman Coulter) were considered. In the ELISA experiments (Fig. 5) the absorbance of two replicas were considered for each of the Nbs. For apparent $K_D$ values of the nanobodies (x$10^{-9}$ M), the median fluorescence intensity values by cytometry and the absorbance values by ELISA were analyzed to obtain best-fit values, standard errors and 95% confidence intervals (Table 1) according to "One site—Specific binding" non-linear least squares regression method in Prism 6 software (Version 6.05, GraphPad Software, Inc.). Curves in Figs. 4b, d and 5 were fitted according to non-linear least squares regression methods in Prism 6 software (Version 6.05, GraphPad Software, Inc.). For each of the fractions (unbound, bound) in Supplementary Fig. S3A, colonies of plated dilutions were counted in each of two independent experimental replicas and then used to calculate the mean values, respectively. Totals, percentages, means and standard deviation values to represent Supplementary Fig. 3SA-C were calculated using Microsoft Excel. Supplementary Fig. S3A and Supplementary Fig. S3B were represented using Prism 6 software (Version 6.05, GraphPad Software, Inc.), while Supplementary Fig. S3C was represented using Microsoft Excel.

**Reporting summary**. Further information on research design is available in the Nature Research Reporting Summary linked to this article.

## Data and materials availability

Source data for the graphs and charts in the main figures are available as Supplementary Data 1 and any remaining information can be obtained from the corresponding author upon reasonable request. Plasmids, strains and other constructs are also available upon request.

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

## Acknowledgements

This work was funded by the SETH (RTI2018-095584-B-C42) (MINECO/FEDER) and SyCoLiM (ERA-COBIOTECH 2018 - PCI2019-111859-2) Projects of the Spanish Ministry of Science and Innovation, the MADONNA (H2020-FET-OPEN-RIA-2017-1-766975), BioRoboost (H2020-NMBP-BIO-CSA-2018-820699), SynBio4Flav (H2020-NMBP-TR-IND/H2020-NMBP-BIO-2018-814650) and MIX-UP (MIX-UP H2020-BIO-CN-2019-870294) Contracts of the European Union and the InGEMICS-CM (S2017/BMD-3691) Project of the Comunidad de Madrid - European Structural and Investment Funds (FSE, FECER). LAF is supported by Grant BIO2017-89081-R from *Agencia Española de Investigación* (AEI/MICIU/FEDER, EU), PIE-RDL-COVID-19 (CSIC, MICIU) and the European Union's Horizon 2020 Future and Emerging Technologies research and innovation programme (Grant: FET Open 965018- BIOCELLPHE).

## Author contributions

V.dL., L.A.F. and A.N. designed the research. Y.A.R. and Y.M. performed the research. L.C. provided technical assistance. C.P. and G.F. supported the work throughout. A.N. and L.A.F. analyzed data. Y.A.R. and V.dL. wrote the original draft of this paper and prepared the figures. All authors read and approved the final manuscript.

## Competing interests

A.N. and C.P. are inventors on a patent related to directed evolution with random genomic mutations (DIvERGE) (US10669537B2 and WO2018108987: Mutagenizing Intracellular Nucleic Acids). All other authors declare no competing interests.
