## [Peer Review File · Communications Biology]

ssDNA recombineering boosts in vivo evolution of nanobodies displayed on bacterial surfacesReviewers' comments:

Reviewer #1 (Remarks to the Author):

Review:

Yamal Al-ramahi et al. describe an accelerated laboratory evolution method that applies ssDNA recombineering and bacterial surface display to engineer nanobody epitope recognition. Engineering nanobody recognition is important to a wide range of applications, that include nanobody therapies, diagnostics, and drug delivery. To highlight this method's utility, Yamal Al-ramahi et al. evolved TD4 recognition to bind the translocated intimin receptor (TirM) of EPEC pathogens, which TD4 natively binds EHEC TirM (70% epitope homology). Using FACS and ELISA, the authors discovered TD4 can be evolved to bind EPEC TirM, while maintaining specificity towards EHEC TirM. Even though there are many available methods for engineering antibody recognition (ribosomal-, phage-, bacteria-, and yeast-display), this new method by-passes the need for laborious cloning and mutagenesis procedures. In addition, TirM is a therapeutically important target because it is required for bacterial pathogenesis, specifically binding to intestinal epithelia cells, therefore, this work is impactful and broadly important.

General Comments:

The authors claim that one of the main advantages of their method is that it generates more diversity than traditional protein engineering approaches, yet the authors never quantify the diversity of their method nor benchmark their method to other approaches. Even though library diversity is an important parameter for directed evolution, I believe a major advantage of this method is the type/breadth of diversity one can achieve. For example, to obtain the same type of targeted mutagenesis across an expanded region (90 nts) using degenerate primers would require numerous primer pairs. In addition, error prone PCR has biases and would be difficult to target three regions. Could the authors please add a few sentences emphasizing this in a little more detail.

Specific Comments:

- Line 70, what is the diversity of the natural immune system, and what is the diversity of the mentioned methods?
- Line 74, expand discussion/advantages to include the type and breadth of diversification (see general comment).
- Line 92, not clear how antibody engineering can apply to environmental applications. Please explain.
- Line 110, "to target antigens by". By what? Please clarify. What structures are the authors referring to?
- Line 112, how does the engineered strain with modified surface structures apply to in solution?
- Line 123, improving recombination by how much?
- Line 126, Please define the criteria?
- Line 127, why ten rounds? Wouldn't subsequent rounds erase the previous rounds mutations? If mutating each CDR in a single library, I can see why three rounds would be necessary, but I do not see why more than 10 is necessary. Also, based on description, it isn't clear whether three distinct libraries were made where each CDR region is mutated independently, or if iterative mutations were made to each CDR region, thereby diversifying all regions in one overall library.
- Figure 2,
 - o What is y axis, actual counts or thousands of counts?
 - o B-C: Highlight codon and nucleotide region being mutated
 - o C,D: use same X-axis.
 - o Quantify library diversity using a rarefaction curve.
 - o D: Not clear whether diversity observed is NGS error or nonspecific mutagenesis from pORTMAGE3 plasmid. Include control of no pORTMAGE3 plasmid and no ssDNA transformation.
- Line 132, what is diversity? You can do rarefaction curve analysis: Please refer to Figure S2, Cell

2019 Oct 3;179(2):459-469.e9. doi: 10.1016/j.cell.2019.09.015.

- Line 135, include explanation from figure caption: "The lower mutagenic efficiency of the oligonucleotide pool targeting CDR1 may have resulted from a higher tendency to form secondary structures which could hinder the transformation process or the incorporation of these oligonucleotides during recombineering."
- Line 136, how can region 1 be counter selected in vivo. I do not see what selection pressure would cause this.
- Line 137, regions mentioned are not highlighted in figure.
- Line 139, discuss control experiment of no ssDNA transfection. Also, please include additional control experiment of no pORTMAGE3 control.
- Combine Figures 3-5
- Line 149, include control of E. coli libraries of no ssDNA transfection or no pORTMAGE3. It isn't clear whether the mutants obtained are natural mutants or from evolution method.
- Line 153, after 5 rounds, why is there such a large % of WT if WT has no affinity?
- Figure 4a, why does the fluorescent population digress in the fifth round. Does this mean the affinity has decreased? Why is there a 50/50 population in the second round if WT has no affinity?
- Line 170, what sequencing was used, NGS or PacBio?
- Line 178-179, does SNP make sense based on # of rounds of mutations (10) and mutating each region? Maybe, libraries were made independently, but this isn't clear?
- Line 192 and Figure 6C, the authors mention similar binding, but the data shows that the evolved variants bind EHEC TirmC stronger than WT TD4.
- Figure 6B and D, label Kd and error for each curve.
- Line 205, define Kd exactly and include error for each variant. Also, I could not find Table 1 in main text or supplemental.
- Line 210, I do not understand how this claim can be substantiated based on the data. Seems speculative.
- Line 220, (Methods section:-> (Methods Section)
- Line 223, (Vamy-> (Vamy)
- Line 226, include STDEV for Kd
- Line 229, include STDEV for Kd
- Line 230, please explain further, as in, what are the affects (kon or koff?), and why aren't these affects consistent amongst the variants. Could perform SPR or BLI to determine kon and koff.
- Line 232, wouldn't this affect be the same for all variants?
- Line 246, can you model which TirM residues the nanobody variants (mutant AA) interact with directly? If so, this would be a nice discussion point.
- Line 254-255, how does this compare to FACS data. Please include this discussion.
- Line 265, please define range.
- Line 266, what are the different needs to have varying affinities. Please explain further.

Reviewer #2 (Remarks to the Author):

This contribution describes an example of changing the selectivity of a nanobody for a new target while maintaining high affinity to the original target using recombineering in bacteria. The presentation of nanobodies is the same as described before, using intimin, while recombineering targeting CDRs is able to strongly increase the affinity in few rounds of selection, while maintaining the affinity for the initial target.

Conclusions of the paper are justified based on experimental results, methods are well described and of interest to other researchers, for the generation of new nanobodies but also for selection of other protein binders.

It would be useful to expand in the discussion the connection between maintaining the affinity against the initial target and the selection of the mutagenic primers with 98.5% of the ratio of the initial nucleotide. On the other hand limitations of this method for the generation of nanobody variability such as e.g. lack of deletions and insertions could be added.

Reviewer #3 (Remarks to the Author):

Al-ramahi et al. describe the use of ssDNA recombineering (i.e. MAGE) to create nanobody libraries with diversified CDR loops in *E. coli*. They carry out 10 cycles of MAGE using a collection of ssDNAs targeted to CDR regions in a parent nanobody, TD4, encoded for surface-display in *E. coli*. The resulting TD4 nanobody library was then subject to a standard MACS enrichment process to find TD4 variants that improve binding for TirM(EPEC) instead of TirM(EHEC). The two antigens are related, but the parent TD4 nanobody only binds the EHEC version well whereas the selected nanobody binds the EHEC version well, proving that they can increase affinity of TD4 for a new target.

My main concern with this manuscript is that the use of MAGE to make nanobody libraries does not seem in my mind to be an improvement over simply making a standard plasmid-encoded nanobody library with diversified CDRs. MAGE is effective when diversification and selection are repeated in cycles where each cycle consists of a diversification step followed by a selection step. By cycling diversification and selection, one can "climb" fitness peaks through the successive accumulation and fixation of beneficial mutations. However, in this manuscript, MAGE is simply used as a nanobody library synthesis tool. The selection comes after the library is already fully generated. To show that MAGE has value for nanobody evolution, the authors need to carry out successive rounds of MAGE followed by MACS followed by MAGE followed by MACS, etc. where we see accumulation of mutations and improvement of fitness over successive MAGE-MACS rounds. Perhaps that is not easy to do because MAGE ssDNA oligos will replace previously fixed mutations in subsequent rounds, but it might be possible if the oligos are designed to be far apart or if there is incomplete copying from a mutagenic oligo. As the manuscript stands, I can't seem to pinpoint an advance here over making a standard plasmid-encoded nanobody library and subjecting it to MACS. In fact, given that the outcomes of the nanobody evolution experiment are single mutants that improve affinity, I would guess that a plasmid-encoded nanobody library with diversified CDRs would find the same, and it should be much easier to make a standard plasmid-encoded nanobody library. I welcome clarification from the authors if I am missing something.

In addition to addressing the above concern, the binding ELISA measurements should be further validated if the authors are to claim K_d values. In the ELISA procedure used, it is unclear whether steady state has been achieved before washing. The authors should demonstrate that steady state binding is reached before wash. Ideally, another method for K_d measurement should be used, for instance SPR, as this type of ELISA can be inaccurate and condition dependent.

Response to Reviewers' comments¹ to COMMSBIO-21-0452-T and how they have been addressed in the now uploaded version (in blue)

Reviewer 1

... this work is impactful and broadly important.

Thanks! We appreciate the thorough review of the ms. and the many useful hints.

General Comments:

The authors claim that one of the main advantages of their method is that it generates more diversity than traditional protein engineering approaches, yet the authors never quantify the diversity of their method nor benchmark their method to other approaches. Even though library diversity is an important parameter for directed evolution, I believe a major advantage of this method is the type/breadth of diversity one can achieve. For example, to obtain the same type of targeted mutagenesis across an expanded region (90 nts) using degenerate primers would require numerous primer pairs. In addition, error prone PCR has biases and would be difficult to target three regions. Could the authors please add a few sentences emphasizing this in a little more detail.

Thanks for the comment. Comparison of alternative method to similar ends can be very slippery if they are not carried out side-by-side. We argue that the main advantage of our approach is the option of running a large number of recombineering cycles on the same strain and the possibility to calibrate mutation frequencies by playing with the composition of the oligonucleotide cocktail. Quantitation of the procedure is described in detail in Nyerges *et al* (2018).

Specific Comments:

- Line 70, what is the diversity of the natural immune system, and what is the diversity of the mentioned methods?

Numbers on this question and adequate references have been added to the revised text.

- Line 74, expand discussion/advantages to include the type and breadth of diversification (see general comment).

Text expanded to the revised Introduction for addressing this question.

- Line 92, not clear how antibody engineering can apply to environmental applications. Please explain.

For example, monitoring of catabolic enzymes in complex samples. A new reference on the matter has been added to the revision

- Line 110, "to target antigens by". By what? Please clarify. What structures are the authors referring to?

Text modified and referenced for clarity.

- Line 112, how does the engineered strain with modified surface structures apply to in solution?

Text modified and referenced for clarity.

¹ Only specific requests/remarks are reproduced in black font.

- Line 123, improving recombination by how much?

>4 orders of magnitude. New reference added.

- Line 126, Please define the criteria?

Explained in the revised text and a new reference added.

- Line 127, why ten rounds? Wouldn't subsequent rounds erase the previous rounds mutations? If mutating each CDR in a single library, I can see why three rounds would be necessary, but I do not see why more than 10 is necessary.

One can plan these experiments in various ways. Our choice was to generate as much diversification as possible at the start. According to published data on MAGE, 10 recombineering cycles seemed to be adequate to reach a high number of nucleotide replacements (up to 10%).

- Also, based on description, it isn't clear whether three distinct libraries were made where each CDR region is mutated independently, or if iterative mutations were made to each CDR region, thereby diversifying all regions in one overall library.

There was only one library with a cocktail of oligos targeting the 3 CDR sequences. In this way we could cover a broader sequence landscape

- Figure 2,

- o What is y axis, actual counts or thousands of counts?

Corrected. NGS reads.

- o B-C: Highlight codon and nucleotide region being mutated

Done on panel C for clarity (too busy a figure if we mark the same in all cases)

- o C,D: use same X-axis.

It is the same with just 1-nt shift, no way it can be mistaken. We advocate to leave it as it is.

- o Quantify library diversity using a rarefaction curve.

We did not monitor mutant frequencies along the cycles, but just the final outcome after 10 mutagenic rounds. As indicated above, based on earlier literature on MAGE (eg. Wang & Church GM. 2011. Meth Enzymol 498:409-426) made us assume confidently that we had the maximum that the procedure could deliver. Please note new Supplementary Figures S1 and S3C, where the breadth of the thereby generated variability can be visualized.

- o D: Not clear whether diversity observed is NGS error or nonspecific mutagenesis from pORTMAGE3 plasmid. Include control of no pORTMAGE3 plasmid and no ssDNA transformation.

This part of the legend has been expanded and explained in more detail to clarify this. The controls (which apply to the technology as a whole, not just the experiments of this article) were covered in detail in the work of the Authors on the basic recombineering pipeline, incl. the virtual lack of off-site mutations (see Nyerges et al. 2016, 2018).

- Line 132, what is diversity? You can do rarefaction curve analysis: Please refer to Figure S2, Cell 2019 Oct 3;179(2):459-469.e9. doi: 10.1016/j.cell.2019.09.015.

See comment above

- Line 135, include explanation from figure caption: “The lower mutagenic efficiency of the oligonucleotide pool targeting CDR1 may have resulted from a higher tendency to form secondary structures which could hinder the transformation process or the incorporation of these oligonucleotides during recombineering.”

The corresponding text has been expanded to consider also the possibility of deleterious misfolding and ensuing detrimental secretion in sequences affected in CDR1. These explanations are hypothetical, we did not pursue the issue further.

- Line 136, how can region 1 be counter selected in vivo. I do not see what selection pressure would cause this.

See before: secretion problems (the nanobody ought to be secreted after all), can be counterselective *in vivo*

- Line 137, regions mentioned are not highlighted in figure.

Indicated now in Fig. 2C

- Line 139, discuss control experiment of no ssDNA transfection. Also, please include additional control experiment of no pORTMAGE3 control.

See above: These controls are secured on the basis of earlier work with the recombineering procedure.

- Combine Figures 3-5

Done!

- Line 149, include control of E. coli libraries of no ssDNA transfection or no pORTMAGE3. It isn't clear whether the mutants obtained are natural mutants or from evolution method.

See above about controls.

- Line 153, after 5 rounds, why is there such a large % of WT if WT has no affinity?

We provide in the revised text a plausible explanation and a reference in support of it. The effect may be caused by low affinity interactions of wild type Nb to TirMEPEC due to avidity effects favored by the multivalent expression of the Nb on the bacterial surface and their stabilization with anti-biotin magnetic beads.

- Figure 4a, why does the fluorescent population digress in the fifth round. Does this mean the affinity has decreased? Why is there a 50/50 population in the second round if WT has no affinity?

The data show that 3 cycles suffice to select a good pool of new binders. We speculate that further rounds may eliminate weak binders—strong ones will remain regardless.

- Line 170, what sequencing was used, NGS or PacBio?

Clarified: PacBio

- Line 178-179, does SNP make sense based on # of rounds of mutations (10) and mutating each region? Maybe, libraries were made independently, but this isn't clear?

See above: a single cocktail of oligos targeting the 3 CDRs were used.

- Line 192 and Figure 6C, the authors mention similar binding, but the data shows that the evolved variants bind EHEC TirM stronger than WT TD4.

The parts of the text dealing with affinities has been thoroughly revised for consistency and clarity

- Figure 6B and D, label Kd and error for each curve.

Done

- Line 205, define Kd exactly and include error for each variant. Also, I could not find Table 1 in main text or supplemental.

Done. Table 1 is in the main text.

- Line 210, I do not understand how this claim can be substantiated based on the data. Seems speculative.

This is indeed a speculation (we just suggest it). But given that the wt sequence has no affinity for the new antigen we can safely argue that the mutations do create a genuinely new interaction.

- Line 220, (Methods section:-> (Methods Section)

Corrected

- Line 223, (Vamy-> (Vamy)

Corrected

- Line 226, include STDEV for Kd

Done

- Line 229, include STDEV for Kd

Done

- Line 230, please explain further, as in, what are the affects (kon or koff?), and why aren't these affects consistent amongst the variants. Could perform SPR or BLI to determine kon and koff.

The method used does not enable a fine analysis of the interaction constants, as we deal with both in vivo (cell-bound) and in vitro (purified proteins) scenarios. Yet, the data enables us to compare gross behavior and they document acquisition of new affinities.

- Line 232, wouldn't this affect be the same for all variants?

It could or not. There is room to argue that each nanobody may behave differently

- Line 246, can you model which TirM residues the nanobody variants (mutant AA) interact with directly? If so, this would be a nice discussion point.

The 3D structure of the TirM^{EHEC} antigen bound to parental Nb TD4 is not available and running a model without these data would be highly speculative.

- Line 254-255, how does this compare to FACS data. Please include this discussion.

Some additional discussion has been added, although we acknowledge that we enter a hypothetical realm.

- Line 265, please define range.

Affinities are summarized in Table 1.

- Line 266, what are the different needs to have varying affinities. Please explain further.

Depending on the specific targets and their levels in a given sample different affinities could be necessary for a successful detection. This is a general scenario and we would prefer not to be more specific.

Reviewer 2

... Conclusions of the paper are justified based on experimental results, methods are well described and of interest to other researchers ...

Thanks!

It would be useful to expand in the discussion the connection between maintaining the affinity against the initial target and the selection of the mutagenic primers with 98.5% of the ration of the initial nucleotide.

In this work we adopt just one ratio of nt changes in the mutagenic oligonucleotides that has been proven to work well in other contexts. But the frequencies and types of mutations can indeed be modulated by playing with such a %.

On the other hand limitations of this method for the generation of nanobody variability such as e.g. lack of deletions and insertions could be added.

There is a whole range of methods to generate diversity in nanobodies. Indeed, the one discussed here with the oligonucleotide mix used fails to generate deletions and insertions, but we see that as an advantage rather than a limitation.

Reviewer 3

... My main concern with this manuscript is that the use of MAGE to make nanobody libraries does not seem in my mind to be an improvement over simply making a standard plasmid-encoded nanobody library with diversified CDRs.

The hereby presented recombineering method is optimal for chromosomally-encoded antibody genes. Multicopy plasmids can enter a considerable noise in the MACS enrichment, since multiple copies of different plasmids would exist and the same cell may express diverse types of surface-exposed nanobodies. To avoid this, transformation of mutated plasmids to fresh cells would be needed, which increases manipulation and preclude performing a continuous genomic diversification *in vivo*. In any case, combining DiVERGE with surface-exposed nanobodies in *E. coli* is our opinion a considerable advance in respect to alternative plasmid-based methods which involve *in vitro* mutation, ligations, transformations, etc.

MAGE is effective when diversification and selection are repeated in cycles where each cycle consists of a diversification step followed by a selection step. By cycling diversification and selection, one can "climb" fitness peaks through the successive accumulation and fixation of beneficial mutations. However, in this manuscript, MAGE is simply used as a nanobody library synthesis tool.

As indicated in one of the responses to Rev #1 above, the diversification/selection protocol can be planned in various ways. In the case reported in this work we indeed focused on generating maximum diversity at the start followed by a selection cycles later. But indeed, the cycles could be alternated intermittently when required for specific purposes, the methodology is the same.

The selection comes after the library is already fully generated. To show that MAGE has value for nanobody evolution, the authors need to carry out successive rounds of MAGE followed by MACS followed by MAGE followed by MACS, etc. where we see accumulation of mutations and improvement of fitness over successive MAGE-MACS rounds.

See comment above.

Perhaps that is not easy to do because MAGE ssDNA oligos will replace previously fixed mutations in subsequent rounds, but it might be possible if the oligos are designed to be far apart or if there is incomplete copying from a mutagenic oligo.

The alternative experiment proposed in indeed feasible and could work well if a strong selective pressure is maintained all along. But in the work presented in this paper we just followed a different workplan—which eventually delivered what we were after.

As the manuscript stands, I can't seem to pinpoint an advance here over making a standard plasmid-encoded nanobody library and subjecting it to MACS. In fact, given that the outcomes of the nanobody evolution experiment are single mutants that improve affinity, I would guess that a plasmid-encoded nanobody library with diversified CDRs would find the same, and it should be much easier to make a standard plasmid-encoded nanobody library. I welcome clarification from the authors if I am missing something.

See above. Plasmids-based nanobody variant libraries are problematic for this type of applications. And *in vitro* insertions of sequence libraries to generate variability is always limited by ligation and transformation.

In addition to addressing the above concern, the binding ELISA measurements should be further validated if the authors are to claim K_d values. In the ELISA procedure used, it is unclear whether steady state has been achieved before washing. The authors should demonstrate that steady state binding is reached before wash. Ideally, another method for K_d measurement should be used, for instance SPR, as this type of ELISA can be inaccurate and condition dependent.

We acknowledge that the methods adopted in this work to inspect affinities were quite unsophisticated. But the main objective was to accredit the emergence of new, factual binding abilities both in an *in vivo* context (cell-bound) and *in vitro* (purified proteins), not a deep analysis of the interactions.

=====

REVIEWERS' COMMENTS:

Reviewer #1 (Remarks to the Author):

Overall Comment: Based on the authors edits, I believe that the revised manuscript is improved. However, there are still some points that were not addressed and need further clarification.

o It still isn't clear why 10 rounds are necessary. My initial understanding was that subsequent rounds would erase the previous round's mutations. However, if recombination efficiency is low, then this would be of little concern. I believe a brief discussion about the recombination efficiency and the probability for erasing previous round's mutations would be beneficial to readers. Especially readers new to this technology.

o The authors did include a new SI figure describing mutation frequency, however, they still have not added an analysis discussing library diversity. Based on the available data, it seems that this analysis is possible. Please discuss the theoretical diversity and experimental diversity achieved.

o Could the authors explain the difference between the PacBio (Figure S1) and Illumina sequencing (Figure 2) data? Based on PacBio data, the mutation frequency is ~0.15%, which equates to WT being ~600 fold excess. However, in Figure 2, the authors observe mutations in the CDR regions as most prominent (~3x). The Illumina result would make sense if WT was subtracted, but all 4 nt are observed? I apologize if I am misinterpreting these results.

o I still do not understand the negative selection argument regarding region CD1. This would make sense if the displayed protein or intimin display system was an essential gene, but it isn't, and there is not a selection pressure selecting for it. I believe DNA secondary structure or reduced recombination is best explanation.

o I leave the disagreement about the discrepancy between the X-axis of Figure 2C and 2D up to the Journal editor. I believe Nature Communication has high standards and that the axis should be corrected.

o Overall, the manuscript is much improved, and upon clarifying these points, I believe it is ready for publication.

Reviewer #2 (Remarks to the Author):

Authors have appropriately resolved the issues and the manuscript is appropriate for publication.

Reviewer #3 (Remarks to the Author):

It is still not obvious to me, and not shown in the examples, why ssDNA recombineering is a significant improvement over traditional methods for making nanobody libraries. What the authors show is that ssDNA recombineering can make a large surface displayed nanobody library from which new binders can be selected, but this is something possible with a plasmid library too. It's possible that ssDNA recombineering can result in larger libraries. Ideally, the authors would show that they have in fact generated larger libraries with cycles of ssDNA recombineering and that they are more productive sources of desired binders.

The authors discuss that a plasmid library has more copy number noise and suggest that cells may not be clonal when transformed with a plasmid library (which I disagree with - usually, transformation bottlenecks result in clonality). However, these are not major problems that prevent the success of plasmid-based nanobody libraries.

Overall, I stand by my original review that the experiments do not show an obvious advantage over existing nanobody libraries.

Nonetheless, I don't have any objections to this manuscript being published.

Response to Reviewers' comments¹ to COMMSBIO-21-0452A and how they have been addressed in the revised version (in blue)

Reviewer #1

- It still isn't clear why 10 rounds are necessary. My initial understanding was that subsequent rounds would erase the previous round's mutations. However, if recombination efficiency is low, then this would be of little concern. I believe a brief discussion about the recombination efficiency and the probability for erasing previous round's mutations would be beneficial to readers. Especially readers new to this technology.

Reviewer is right in that subsequent mutagenic rounds increase the overall diversification. On the other hand, the issue of recombination efficiency with these technologies has been tackled in detail in the original descriptions of MAGE (ref 16) and DiVERGE (ref 18). We thus believe that such a discussion can be left aside in the current ms.

- The authors did include a new SI figure describing mutation frequency, however, they still have not added an analysis discussing library diversity. Based on the available data, it seems that this analysis is possible. Please discuss the theoretical diversity and experimental diversity achieved.

Our nanobody library was designed to have on average one point mutation within the 3 CDR regions in total in each nanobody variant. This gives a maximum library size of 214 missense variants. However, as we only observed mutagenesis in the 2nd and 3rd CDR region, the predicted library size was approximately 161 missense variants. Almost all of these variants were observed in our PacBio sequencing data, given the small library size compared to the sequencing depth. As before, the pluses and minuses of the technology—and possible improvements—in terms of library diversity have been tackled in previous publications #16 and #18 and we believe that going through them again does not add much to the outcome of this work.

- Could the authors explain the difference between the PacBio (Figure S1) and Illumina sequencing (Figure 2) data? Based on PacBio data, the mutation frequency is ~0.15%, which equates to WT being ~600 fold excess. However, in Figure 2, the authors observe mutations in the CDR regions as most prominent (~3x). The Illumina result would make sense if WT was subtracted, but all 4 nt are observed? I apologize if I am misinterpreting these results.

Both our PacBio (Figure S1) and Illumina (Figure 2C) results show mutational frequencies per position. While it is true that the wild-type sequences remain a majority, the cumulative ratio of mutants in the entire library is approximately 6.3%. On our Fig 2C, we did not subtract the WT mutation rate in order to show the high background sequencing noise. In fact, we included Fig 2D that shows the background. In Fig. 2C and D, every bar at every nucleotide position of the nanobody sequence shows the observed frequency of the 3 possible mismatching nucleotides. Due to its lower error rate, the PacBio data seems to be more reliable for assessing library diversity.

- I still do not understand the negative selection argument regarding region CD1. This would make sense if the displayed protein or intimin display system was an essential gene, but it isn't, and there is not a selection pressure selecting for it. I believe DNA secondary structure or reduced recombination is best explanation.

Explanations for the lack of mutants in CD1 are all hypothetical. One plausible scenario is that misfolded proteins may clog secretion systems and cause toxicity. Yet, we do mention the additional possibilities suggested by this reviewer in the revised version.

¹ Only specific requests/remarks are reproduced in black font.

• I leave the disagreement about the discrepancy between the X-axis of Figure 2C and 2D up to the Journal editor. I believe Nature Communication has high standards and that the axis should be corrected.

Fig. 2 has been corrected to align the X-axes of panels 2C and 2D, as requested.

• Overall, the manuscript is much improved, and upon clarifying these points, I believe it is ready for publication.

Thank you!

Reviewer #2

No remaining issues, thank you!

Reviewer #3

It is still not obvious to me, and not shown in the examples, why ssDNA recombineering is a significant improvement over traditional methods for making nanobody libraries. What the authors show is that ssDNA recombineering can make a large surface displayed nanobody library from which new binders can be selected, but this is something possible with a plasmid library too. It's possible that ssDNA recombineering can result in larger libraries. Ideally, the authors would show that they have in fact generated larger libraries with cycles of ssDNA recombineering and that they are more productive sources of desired binders.

Claiming that one technology is superior to others would require side-to-side comparisons that are by no means the objective of this paper. Yet, we believe this work adds one more, conceptually original and quite useful tool to the battery of methods for generating and evolving recombinant nanobodies in bacterial platforms.

The authors discuss that a plasmid library has more copy number noise and suggest that cells may not be clonal when transformed with a plasmid library (which I disagree with - usually, transformation bottlenecks result in clonality). However, these are not major problems that prevent the success of plasmid-based nanobody libraries.

Plasmid-based libraries are indeed feasible also—we have used them successfully eg. doi: 10.1021/acssynbio.9b00375—but we still argue that for the specific application described in this work (generation of new nanobodies), having the target gene in the chromosome makes expression less noisy.

Overall, I stand by my original review that the experiments do not show an obvious advantage over existing nanobody libraries.

See above

Nonetheless, I don't have any objections to this manuscript being published.

Fair enough, thanks!

=====